# Using the Proteomics Toolbox to Resolve Topology and Dynamics of Compartmentalized cAMP Signaling

**DOI:** 10.3390/ijms24054667

**Published:** 2023-02-28

**Authors:** Duangnapa Kovanich, Teck Yew Low, Manuela Zaccolo

**Affiliations:** 1Center for Vaccine Development, Institute of Molecular Biosciences, Mahidol University, Nakhon Pathom 73170, Thailand; 2UKM Medical Molecular Biology Institute (UMBI), Universiti Kebangsaan Malaysia, Kuala Lumpur 56000, Malaysia; 3Department of Physiology, Anatomy and Genetics and Oxford NIHR Biomedical Research Centre, University of Oxford, Oxford OX1 3PT, UK

**Keywords:** cAMP signaling, cAMP compartmentalization, G-protein coupled receptor, A-kinase anchoring protein, phosphodiesterases, protein kinase A, proteomics

## Abstract

cAMP is a second messenger that regulates a myriad of cellular functions in response to multiple extracellular stimuli. New developments in the field have provided exciting insights into how cAMP utilizes compartmentalization to ensure specificity when the message conveyed to the cell by an extracellular stimulus is translated into the appropriate functional outcome. cAMP compartmentalization relies on the formation of local signaling domains where the subset of cAMP signaling effectors, regulators and targets involved in a specific cellular response cluster together. These domains are dynamic in nature and underpin the exacting spatiotemporal regulation of cAMP signaling. In this review, we focus on how the proteomics toolbox can be utilized to identify the molecular components of these domains and to define the dynamic cellular cAMP signaling landscape. From a therapeutic perspective, compiling data on compartmentalized cAMP signaling in physiological and pathological conditions will help define the signaling events underlying disease and may reveal domain-specific targets for the development of precision medicine interventions.

## 1. Introduction

cAMP is a ubiquitous secondary messenger. In eukaryotes, cAMP production is triggered by the activation of G protein-coupled receptors (GPCRs), a large and diverse protein family with around 800 members [1] that transmit signals from a variety of stimuli (e.g., hormones, cytokines, neurotransmitters, mechanical stress). Upon activation, the receptor undergoes a conformational change to achieve its active state, followed by coupling to and activation of heterotrimeric G-proteins (α, β and γ). This initiates a signaling cascade that triggers adenylyl cyclase (AC) to catalyze the conversion of ATP to cAMP. Elevated intracellular cAMP then activates a small number of effector proteins, including the cAMP-dependent protein kinase (PKA), the exchange factors activated by cAMP (EPAC), the hyperpolarization-activated cyclic nucleotide-gated channels (HCN) and the popeye domain-containing proteins (POPDC). Despite relying on a limited number of effectors, cAMP signaling is central to multiple physiological processes, ranging from proliferation, differentiation, metabolism, and control of specialized cellular activities, such as synaptic transmission, cardiac contraction and hormone secretion. Unsurprisingly, disruption of cAMP signaling is associated with several diseases, including heart failure [2] and cancer [3].

The most extensively studied target of cAMP is PKA. The inactive PKA holoenzyme consists of two regulatory (R) and two catalytic (C) subunits. There are four versions of R subunits, further classified as type I (RIα, RIβ) and type II (RIIα, and RIIβ), and three versions of C: Cα, Cβ, and Cγ. cAMP binds to R in the inactive tetrameric holoenzyme R2C2 inducing a conformational change that removes the inhibitory action of R on C subunits, leading to phosphorylation of nearby substrates. PKA is an extremely promiscuous kinase with the ability to phosphorylate many targets within the same cell, thus contributing to the ability of cAMP to mediate a multiplicity of cellular effects. It is now widely accepted that cAMP signaling achieves specificity of response via compartmentalization. Given the large number and variety of activating stimuli, receptors and PKA substrates, cAMP/PKA signaling is organized within the cell as a collection of multiple pathways controlled by a precise spatial and temporal coordination of signal transduction. Such organization enables the correct translation of the message conveyed by a given external stimulus into the appropriate phosphorylation events and cellular responses [4].

The spatial constraint on cAMP signaling became obvious with the observation, almost 40 years ago, that stimulations of cardiac myocytes with isoproterenol or prostaglandin E1 led to different cellular responses despite the fact that a similar increase in cAMP was generated in response to the two stimuli [5]. Evidence supporting the concept of PKA compartmentalization came later, with the identification of several scaffolding proteins, termed A-kinase anchoring proteins (AKAPs), that tether PKA at specific subcellular locations [6,7]. Currently, the AKAP family includes around 60 diverse proteins which share a characteristic R-binding domain [8]. It is now recognized that AKAPs coordinate unique cAMP signaling domains, or signalosomes, by recruiting and anchoring PKA in proximity to its substrates and by agglomerating other signaling components, including additional kinases, phosphatases, ACs, and the cAMP-hydrolyzing phosphodiesterases (PDEs), to form signaling hubs with distinct subcellular localization [8]. Specific anchored pools of PKA can then be selectively activated by a spatially confined pool of cAMP generated in response to receptor activation. The active kinase subsets, in turn, phosphorylate target proteins anchored in proximity, avoiding crosstalk and signal contamination among different cAMP/PKA signaling hubs.

One example of cAMP signaling compartmentation is the organization of signaling domains at the plasma membrane, where signaling domains are organized by scaffolding proteins that may bring together GPCRs, cAMP effectors, cAMP producing (ACs) and degrading (PDEs) enzymes, signaling regulators and targets into functional domains that are few tens of nanometers in size [9,10,11] (Figure 1). The protein components (proteome) of each signalosome can be very dynamic, consisting of constitutive as well as contextually recruited or dissociated proteins across activation states and in physiological versus pathological conditions [12,13,14]. The spatiotemporal dynamics of the cAMP/PKA signalosomes make the characterization of such signaling domains daunting.

Although GPCRs are one of the most successful therapeutic target families, current drug discovery programs remain associated with very high attrition rates due to the complexity of the signaling associated with these receptors. The realization that cAMP signaling is organized in discrete subcellular domains offers a new opportunity for the development of drugs that selectively target individual domains for therapeutic interventions that can achieve subcellular precision. Establishing the blueprint of individual cAMP signaling compartments across cell types, activation states and disease states would provide invaluable insight for targeted drug design. Quantitative proteomics methodologies that uncover protein phosphorylation, interaction and proximity provide powerful tools for this type of analysis and are increasingly being applied to study cAMP compartmentalization. Basic principles of the methods and specific examples of the applications to address particular questions are described in detail below.

## 2. The Proteomics Toolbox

cAMP signaling uses different mechanisms, including protein–protein interactions (PPIs) and protein phosphorylation, to relay, process, and translate signals into cellular responses. cAMP signaling compartmentalization heavily relies on the formation of local signaling domains where cAMP signaling components involved in a specific cellular response cluster together. Compartmentalization is critically dependent on the assembly of multiple signaling components that come together via protein–protein interactions to become functional signaling units. Within such domains, cAMP signals are translated into specific cellular responses via the phosphorylation of target proteins. As such, mapping the domain interaction landscape and defining the downstream phosphorylation events are the key aspects of compartmentalized signaling studies. Typically, the proteomics analysis of PPIs and protein phosphorylation is usually performed in a quantitative manner, so the changes in phosphorylation events across experimental conditions can be compared, and true signalosome components can be distinguished from the background. To guide the reader through the applications discussed in this review paper, in Box 1, we describe the general principles of quantitative proteomics.

Box 1Quantitative proteomics in a nutshell.Currently, liquid chromatography-tandem mass spectrometry (LC-MS/MS) is the main workhorse driving proteomics studies. In bottom-up proteomics, proteins are first extracted and proteolyzed with a protease, such as trypsin, to yield smaller peptides [15]. To reduce sample complexity, protein/peptide mixtures can be fractionated with gels or by chromatography [16]. During LC-MS/MS analysis, each peptide fraction is separated with a C18 reversed-phase column connected online to a mass spectrometer. C18-separated peptides are eluted and ionized before measuring the intact mass (MS1) and the fragment mass (MS2) of a peptide and its fragments upon gas phase dissociation in the mass spectrometer. Both the intact and fragment masses are subsequently used for matching a corresponding peptide sequence and a protein(s) from a protein database.To enable protein quantification, (i) label-based and (ii) label-free methods can be additionally coupled to LC-MS/MS. In label-based methods, stable isotopes are usually introduced in a protein/peptide via (i) metabolic (in vivo) labeling or (ii) chemical (in vitro) labeling [17,18,19,20,21,22,23]. In SILAC (stable isotope labeling with amino acids in cell culture), cells are cultured in a special cell culture medium containing stable isotope-labeled versions of amino acids, such as “heavy” arginine or lysine, that can be metabolically incorporated in newly synthesized proteins [17,24]. After cell lysis, proteins from differentially labeled populations are combined, avoiding experimental variations that may be introduced by subsequent sample processing. In MS1 spectra, each SILAC-labeled peptide manifests itself as a doublet or triplet with distinct mass differences and quantification is based on the difference in peak intensities of these multiplets (Figure 2A). Originally, SILAC was designed to compare only two or three cell populations as only three isotopically distinct versions of arginine and lysine are commercially available. Recently, NeuCode SILAC, which enables simultaneous comparison of up to nine treatments and control (18-plex), was developed [25].It is noteworthy that not all biological samples are amenable to metabolic labeling, notably clinical samples. For these samples, chemical labeling, which involves in vitro reaction between labeling reagents and proteins/peptides, is applicable to any biological samples with the caveat of higher experimental variations since labeled samples are mixed at a later step. Chemical labeling reagents contain different heavy isotopes to produce mass shifts in the MS1 spectrum (e.g., dimethyl labeling [21,26]) or MS2 spectrum (e.g., isobaric labeling [22,23]). Dimethyl labeling was developed as a cost-effective platform that is performed at the peptide level (after proteolysis) and involves reductive dimethylation of all primary amines (N-terminus of peptides and ε-amino group of lysine) with isotopomeric dimethyl labels. Similar to SILAC, co-eluting dimethyl-labeled peptides manifest themselves as doublets or triplets with distinct mass differences in MS1 spectra and quantification is performed by comparing the differences in the ion intensity of the differentially labeled peptides (Figure 2A) [21,26]. For studies that require higher level of multiplexing, isobaric-tag-based methods, such as iTRAQ and TMT, can be used [22,23]. The isobaric tagging strategy is based on the covalent modification of the primary amines of peptides with different isobaric tags. Peptides originating from each experimental condition are labeled with each isobaric tag in parallel and combined for MS analysis. In contrast to dimethyl labeling, differentially tag-labeled peptides tend to comigrate and appear as a single peak in the MS1 spectra (Figure 2A). However, post peptide fragmentation, each isobaric tag generates a reporter ion with a distinct mass in the MS2 spectra, which is used for relative quantification (Figure 2A). Currently, up to 8-plex iTRAQ and 16-plex TMT reagents are available commercially [27,28], albeit the cost of the labeling reagents is high.For label-free strategies there is no limitation with respect to the number and origin of samples. This approach is therefore, cost-effective and convenient, as no labeling is required. Since samples are separately collected, processed, and mass-analyzed, such experiments need very careful execution to minimize experimental or analytical variations. Label-free quantification can be as simple as spectral counting, where the number of peptide-spectrum matches (PSM) of a protein is used as a proxy for protein abundance. However, spectral counting is biased towards larger proteins, which generate more peptides than small proteins [29]. Recently, several computational platforms, including the MaxLFQ algorithm (embedded in MaxQuant), or the Minora tool (embedded in Proteome discoverer), were developed for more accurate label-free quantification (LFQ) [30,31]. In the LFQ workflow, the peak intensity, or the area under the curve of a peptide ion, is used for relative quantification across samples (Figure 2B) [30,31].Until now, most discovery-based quantitative proteomics studies have been performed with mass spectrometers that operate in a data-dependent acquisition (DDA) mode. In DDA, the top N (usually 10–20) most intense peptide precursors in a survey MS1 scan are sequentially selected for fragmentation and MS2 detection. The downside of this approach is that DDA is biased towards measuring highly abundant proteins. Additionally, due to the stochastic nature of precursor selection, if the number of precursor ions exceeds the number of top N, peptides may not be consistently detected in all LC-MS/MS runs, resulting in many missing values [32,33]. As a solution, data-independent acquisition (DIA) schemes have been implemented. These include SWATH-MS and Boxcar, developed by the Aebersold’s and Mann’s laboratories, respectively [34,35]. In DIA, essentially all peptide precursors in a fixed m/z isolation window are fragmented in parallel and analyzed, resulting in a near complete recording of all MS2 scans and highly reproducible label-free quantification. DIA enables large-scale protein quantification with low variation and high reproducibility, as demonstrated in a multi-laboratory evaluation study [36]. In term of quantification reproducibility, DIA is superior to DDA and can overcome the issue of abundant peptides dominating the MS1 scan [36].

**Figure 2 ijms-24-04667-f002:**
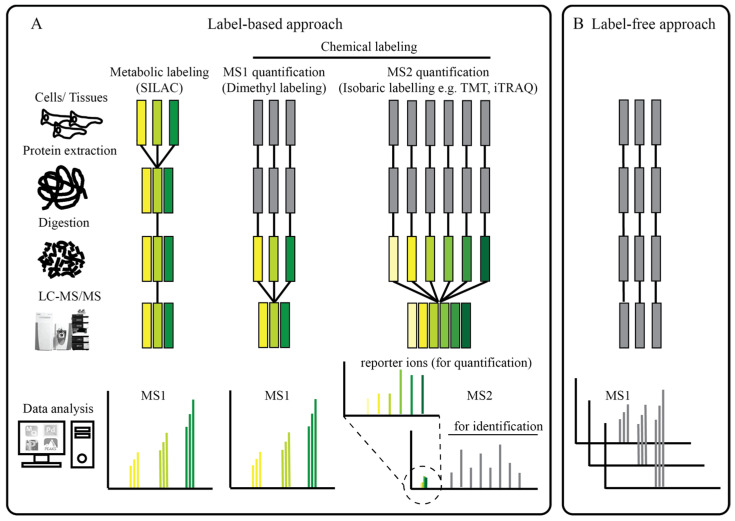
General depiction of various strategies in quantitative proteomics. (**A**) Label-based approaches include the incorporation of stable isotope labels at different steps of sample preparation. The labeled samples are mixed in an equivalent ratio and analyzed in a single LC-MS/MS run. For SILAC and dimethyl labeling, the quantitation is performed by comparing the differences in the ion intensity of the labeled peptide pairs. For the isobaric labeling approach, the quantitation is based on the comparison of reporter ion intensities. (**B**) In the label-free approach, samples are separately prepared and subjected to individual LC-MS/MS runs. Quantification is achieved by run-to-run comparison of the integrated ion intensity or spectral counting.

### 2.1. Protein Interaction and Proximity Profiling—Basic Principles

In MS-based interaction proteomics, the constituents of protein signaling complexes are systematically mapped. Several MS-based interactomics strategies, including affinity purification-MS (AP-MS), proximity labeling-MS (PL-MS), crosslinking-MS and coFractionation-MS (coFrac-MS), have been comprehensively reviewed recently [37]. Among these methods, AP-MS and PL-MS are the most commonly adopted for PPI studies. The AP-MS workflows involve multiple steps, including (i) cell lysis, (ii) incubation of lysate with a specific antibody, followed by capturing with protein A/G beads or with resins conjugated with epitope tag-specific antibodies, (iii) several extensive washing steps to remove non-specific binding proteins and elution of the enriched proteins, and (iv) identification of the eluted proteins by LC-MS/MS. Since purification is performed post-lysis, one challenge is that some detected PPIs may be spurious and non-physiological as the bait and prey may be brought together by chance upon lysis, giving rise to biological false positives. Technical false positives often arise from nonspecific protein binding to the affinity matrices. In addition, mapping the interaction landscape of integral membrane proteins remains a technical challenge. Detergents used to solubilize membrane proteins may disrupt antigen-antibody recognition and membrane protein complexes [38]. Additionally, the success of AP-MS often depends on stable protein interactions, while weak and transient interactors can be lost easily during cell lysis and extensive washing steps.

In recent years, PL-MS has been developed to overcome these limitations. Since its introduction in 2012 [39], PL-MS has been used for PPI mapping and proximity profiling in several cell models and organisms. The technique is based on the labeling of the “neighboring” proteins to the bait protein. These include proteins that physically interact with the bait and other proteins that are in close proximity to the bait. In this approach, the bait protein is expressed as a fusion protein with an enzyme capable of biotin labeling—either as an exogeneous protein or endogenously, under the control of the target gene promoter, using CRISPR/Cas9-mediated genome editing technology [40,41]. Biotin is then added, followed by its catalysis by the fused enzyme into reactive biotin intermediates. These intermediates subsequently diffuse away to biotinylate proteins in the vicinity, a process named promiscuous biotinylation. The labeling strength of these intermediates is limited by the distance of the prey away from the bait. This gives rise to an important concept coined as the “effective labeling radius”. The same experimental settings, but with the expression of free enzyme, are commonly used as a control to reveal false positives that are randomly labeled or that are labeled due to their non-specific association with the enzyme.

If transient interactions have ceased and the binding partners have moved away from the bait protein at the time of extraction, that is not a problem since interacting, and other relevant proteins have already been “marked”. The denaturing condition can then be applied to solubilize the whole proteome without the need to preserve protein interactions. This characteristic of PL approaches is a great advantage for spatiotemporal proteomics. Biotinylated proteins are then purified using avidin/streptavidin-coated beads and are identified by MS. As the biotin-avidin association is the strongest known non-covalent interaction, multiple stringent washing steps can be performed to minimize background contaminants. Similar to AP-MS, to distinguish bona fide proximal proteins from irrelevant labeled proteins, the PL experiment is usually combined with quantitative proteomics approaches, either based on isotopic labeling or label-free quantification.

There are two main enzyme systems used in PL-MS, which are promiscuous biotin ligases, or BioID, in the case of proximity-dependent biotin identification and peroxidases, or APEX, in the case of peroxidase-catalyzed proximity labeling (Figure 3). The promiscuous biotin ligase catalyzes the conversion of biotins to the highly reactive biotinoyl-5′-AMP intermediates to react with proximal primary amines (i.e., lysine residues), leading to the promiscuous biotinylation of surrounding proteins within an estimated radius of 10 nm [42]. Depending on the types of promiscuous biotin ligases used, proximity labeling can occur within ten minutes and up to 18–24 h after the addition of exogeneous biotin [42,43,44,45,46,47] (Figure 3). APEX utilizes modified soybean ascorbate peroxidase to catalyze the oxidation of biotin-phenol to produce highly reactive biotin phenoxyl radicals, which can react with neighboring electron-rich amino acids such as tyrosine and possibly tryptophan, cysteine, and histidine [48]. The reaction requires H_2_O_2_ treatment in the presence of biotin-phenol to produce the radicals, and labeling can be achieved in as short a time as 1 min [48], meaning that the biotin labeling can be timely controlled through H_2_O_2_ availability. This becomes very practical for studies that require cell incubation with ligands for a period of time before initiating the labeling. Additionally, the short half-life of the radicals (<1 ms) and their inability to cross membranes ensures that labeling occurs only within an estimated radius of 20 nm and is contained only within the candidate space [48]. These are, in fact, the key advantages of APEX over BioID. Several current developments in PL-MS have focused on the use of different fusion biotinylating enzymes with a small size to reduce mislocalization and to improve the efficiency and speed of labeling so as to capture PPIs at a higher temporal resolution. These include BioID2 [43], TurboID [44], miniTurbo [44], UltraID [45] and MicroID2 [46] biotin ligases and APEX2 ascorbate peroxidase [49] (Figure 3). The experimental details of PL-MS have been reviewed by several recent publications [37,50,51,52].

### 2.2. Phosphoproteomics—An Overview

cAMP signals are translated into specific cellular responses largely through protein phosphorylation. Accordingly, mapping the cAMP-dependent phosphorylation landscape is key to establishing the topology and functions of cAMP signaling domains. Current state-of-the-art phosphoproteomics technologies allow investigators to identify and quantify at a depth of >10,000 phosphorylation sites in a population of cells in one setting, with high specificity and reproducibility [54,55]. This unprecedented resolution provides a new perspective on the molecular mechanisms underlying diseases and the identification of potential therapeutic targets.

It has been well documented that the identification and quantification of phosphorylated peptides are constrained by their low abundance and stoichiometry relative to their non-phosphorylated counterparts. Besides, during MS data analysis, the proper assignment of a phosphate group among several phosphorylatable residues within a peptide sequence has also posed a considerable challenge [56]. Due to the inherently low stoichiometry, the phosphoproteomics workflows heavily rely on enrichment protocols prior to MS analysis [57]. While the detailed may vary in different workflows, the combination of initial fractionation, phosphopeptide enrichment, stable isotope labeling, and LC-MS/MS has become the method of choice. To enhance the coverage of phosphopeptides, orthogonal peptide fractionation strategies, such as low-pH strong cation exchange (SCX) or high-pH reversed-phase chromatography, are often performed prior to the enrichment [55,58,59,60,61,62]. For quantification, both stable isotope labeling approaches and label-free quantification have been employed [54,55,61,62].

Several affinity enrichment strategies have also been developed to isolate phosphopeptides [57]. For global enrichment of phosphopeptides, immobilized metal affinity chromatography (IMAC) and metal oxide affinity chromatography (MOAC) are the two most popular approaches. IMAC is based on the interaction of positively charged metal ions such as Fe^3+^ or Ti^4+^ with the negatively charged phosphate groups in the phosphopeptides [59,60]. As the name implies, MOAC uses metal oxides, such as TiO_2_, to capture phosphopeptides through the formation of the bidentate binding mode of phosphates to the metal oxide surface [63]. In terms of enrichment efficiency, the two methods are generally comparable, with each showing preferences toward distinct phosphopeptide subpopulations [64,65]. This is not surprising as the two enrichment techniques bind differently to the phosphopeptides. Of note, we have found that the SCX-Ti^4+^-IMAC enrichment method, developed by Heck’s laboratory, shows advantages when it comes to the identification of PKA phosphorylation events [60]. Phosphorylation sites of basophilic kinases, such as PKA or PKC, usually display high basic residue (R/K) content in the neighborhood that can hinder the enrichment of the peptides. The SCX-Ti^4+^-IMAC method was shown to specifically enrich this subset of phosphopeptides [60], therefore, providing extra benefit in the analysis of PKA substrates. Additionally, a more targeted enrichment platform for PKA phosphorylated substrate identification was developed by the same laboratory [66]. The method utilizes specific antibodies against the PKA phosphorylation consensus to capture peptides bearing the motif from cell lysate digests. The high specificity and selectivity of the platform were illustrated by the fact that 98% of the phosphopeptides identified in the study were found to harbor the PKA consensus motif [66].

Phosphoproteomics data analysis involves the identification and quantification of phosphopeptides and the localization of the phosphorylation sites. Even though peptides may be tentatively identified as phosphorylated, it might not be possible to assign the actual sites of modification. In fact, the correct localization of phosphorylation sites is a critical aspect of phosphoproteomic data analysis. Site localization can be complicated when multiple potential phosphoresidues are present within a single peptide, and peptides harboring adjacent or multiple phosphorylated residues can become problematic. In order to resolve the ambiguity between multiple potential sites, the site-determining fragments ions exclusive to a specific site location must be obtained in MS/MS spectra. If these fragment ions are not generated efficiently during fragmentation, the site localization will be significantly impaired. For this purpose, several peptide fragmentation modes have been developed to produce good-quality fragmentation spectra for the assignment of phosphorylation site(s) and are described elsewhere [67]. For a large-scale proteomics experiment involving thousands of phosphopeptides, probability-based site-localization algorithms have been developed and implemented in the main software tools to determine the most likely phosphorylation site(s) and are extensively reviewed elsewhere [68]. In terms of quantification, confident site localization is important since only the phosphopeptides in which phosphorylation sites are assigned can be used for quantitation.

To translate a list of differentially regulated phosphorylation events into biological insight, Gene Ontology (GO) enrichment analysis can be performed to deconvolute the biological processes in which the list of proteins harboring the phosphopeptides are involved [69]. Another important bioinformatic task is the identification of the protein kinases/phosphatases responsible for the observed changes in phosphorylation. Generally, the prediction is based on the kinase phosphorylation motif, and several prediction tools are available and extensively reviewed elsewhere [70]. Interestingly, studies using these prediction tools and investigating the cAMP-associated phosphoproteome have found that phosphorylation mediated by several other kinases, in addition to PKA, is upregulated in response to an increase in cAMP levels [71,72]. These studies have also found that many phosphopeptides are downregulated by cAMP. Overall, these observations indicate that cAMP activates complex signaling networks involving other kinases as well as phosphatases. These networks are bound to be relevant for the regulation of cellular function, although we currently have very little understanding of the role that this more extensive and complex wiring of intracellular signaling plays in cell physiology.

## 3. Application of Proteomics Methodologies to Study Compartmentalized cAMP Signaling

### 3.1. G Proteins Coupled Receptors (GPCRs)

It is widely appreciated that not only can different GPCRs trigger different signals in the same cell, but the same GPCR activated by different agonists can transduce distinct cellular responses through receptor compartmentation [73]. An additional level of signaling complexity is provided by receptor internalization. GPCR signaling is terminated via receptor phosphorylation by GPCR kinases (GRKs), which promotes binding of the receptor to the adaptor proteins β-arrestins (βarrs), leading to receptor desensitization. The internalized receptor either undergoes resensitization by dephosphorylation and recycling back to the plasma membrane or is trafficked to lysosomes for degradation. It is now apparent that different ligands can stabilize distinct receptor conformation states that favor coupling to certain effectors, resulting in the selective activation of certain pathways. So, depending on the ligand, the receptor conformational change can lead to the activation of different G proteins [74] or β-arrestin isoforms [75] or can result in exclusive engagement of the G-protein or β-arrestin pathways [76,77], adding significant complexity to GPCR-mediated signaling.

GPCR signaling was initially believed to happen exclusively at the plasma membrane. In the case of GPCR signaling to ACs, it was demonstrated that the cAMP pool generated by the receptor at the plasma membrane could reach only a nanoscale distance and is short-lived due to local PDE activity [9] and rapid receptor internalization. However, mounting evidence indicates that receptor activation by some ligands, such as peptide hormones, can lead to sustained cAMP production after receptor internalization when the receptor is embedded in the membrane of endosomes or after the trafficking of the internalized receptor to the trans-Golgi network (TGN) [78,79]. Through this signaling modality, cAMP can reach distal organellar targets such as the nucleus [79,80]. In addition, several GPCRs primarily reside at the organellar membranes and trigger distinct inward signaling from these locations inside the cell [81]. Notably, the downstream cellular responses triggered by GPCRs at internal membrane sites were found to be distinct from those elicited by receptors at the cell surface.

In the context of specialized cells, the specificity of the downstream response can be ascribed to differences in receptor compartmentalization at the cell surface as well as at intracellular signaling sites. For example, in cardiomyocytes, β1-adrenergic receptors (β1ARs) and the highly homologous β2-adrenergic receptors (β2ARs) bind to norepinephrine and activate G protein/cAMP/PKA signaling. β2ARs are highly enriched in specialized membrane structures, including T-Tubules and caveolae, whereas the homologous β1ARs are distributed across the entire cell membrane [2,82]. While β1AR stimulation results in more diffuse cAMP signals across the entire cell, β2AR activation generates a cAMP pool more narrowly confined to the site of production [2]. At early time points after receptor internalization, β1ARs and β2ARs appear to traffic to distinct endosomal compartments [83]. β2ARs activate G protein signaling from early endosomes, while the endocytic trafficking route for β1ARs appears to also involve the trans-Golgi network [84,85]. In the heart, β1AR signaling regulates chronotropic, inotropic and lusitropic responses through PKA-mediated phosphorylation of multiple proteins involved in Ca^2+^ handling and excitation-contraction coupling, while β2AR signaling induces modest chronotropic and no lusitropic responses. β1AR signaling drives the expression of pro-apoptotic genes while β2AR signaling promotes antiapoptotic signaling and cardiomyocyte survival, possibly through Gαi coupling or activation of β-arrestin-mediated pathways (as reviewed in [86]). In the failing heart, β2ARs were shown to redistribute to the non-tubular membrane and produced more diffuse β2AR cAMP signals [2], which could lead to the loss of their cardioprotective properties.

From the examples above, it is clear that receptor compartmentalization provides well-defined signaling domains that contribute to the specificity of the downstream response. How exactly this is achieved, however, is far from clear. For example, as the trafficking route of a given receptor appears to depend on the specific ligand bound to it, it is conceivable that the unique conformation the receptor adopts after binding could facilitate the recruitment of distinct compartment-specific sorting proteins that are not presently known. How do cAMP signals generated at internal membrane sites reach the nucleus and selectively drive ligand-specific cAMP-dependent transcription events? The identification of proteins recruited to the internal membrane domains could provide critical cues.

Despite the open questions, recent findings indicate that targeting disease-specific signaling at defined subcellular locations may be a valid alternative to the prevalent and rather blunt approach that targets the receptor at the plasma membrane [87,88]. Such a strategy has the potential to provide more specific therapeutic interventions with reduced undesirable side effects. However, efforts to develop subpathway- or location-specific treatments are presently hindered by our limited understanding of the full makeup of a given receptor signaling network. Elucidation of the complexity of receptor dynamics, as well as of the architecture of signaling complexes with the subcellular resolution, is, therefore, paramount.

APEX-based proximity labeling has been exploited to capture both location- and time-dependency of ligand-specific GPCR signaling dynamics in simplified cell models [89,90,91,92]. One of the advantages of APEX for mapping the GPCR signaling domain is that this rapid labeling technique provides a snapshot that captures the transient signaling landscape within the timescale of GPCR activation (Figure 4) [89,90,91,92]. When combined with a well-designed control system and quantitative proteomics, GPCR-APEX allows identification, in an unbiased manner, of the molecular components in the local environment, including interacting proteins and proximal bystanders to the receptors, through the quantification of thousands of biotinylated proteins. The bystanders identified are considered to store important information on localization and can be used as spatial markers to map the trafficking route of the receptor [89,90,91,92].

As a proof-of-concept, Lobingier et al. investigated the signaling networks associated with the well-studied β2AR and the less explored delta-opioid receptor (DOR) in the HEK293 model system stably expressing fusions of each receptor with APEX2 [89]. Several spatial references were generated, including APEX2-tagged Lyn11 as a plasma membrane marker (PM-APEX2), 2xFYVE-tagged APEX2 as an early endosome marker (Endo-APEX2) and GFP-APEX2 (Cyto-APEX2) as a cytoplasmic reference. Technically, the proteins captured by the APEX labeling method represent a complex mixture of two sets of proteins, the receptor signaling network components and proximal bystanders. To distinguish bystanders from bona fide signaling network components, the abundance of proteins enriched from receptor APEX can be compared to the proteins enriched in the localization-matched control. Bystander proteins from the same subcellular compartment should be equally biotinylated, while the signaling network components should be enriched in the receptor APEX compared sample compared to the control. For instance, by comparing the proteins identified from β2AR APEX with control APEX using PM-APEX2 after 1-min activation, plasma membrane proteins were identified with similar abundance while β-arrestin 2 and proteins of the endocytic adaptor complex AP2, which are recruited to the active receptor, were more enriched in the β2AR-APEX dataset. Similarly, by comparing proteins identified from β2AR APEX with control APEX using Endo-APEX2 10 min past activation, several proteins engaged in β2AR endosomal sorting, including retromer complex components, were enriched while other early endosomal proteins were equally detected, highlighting the importance of selecting appropriate spatial references for the success of the GPCR APEX study [89].

The same strategy was then utilized to study the underexplored DOR signaling and trafficking network, with a focus on the characterization of DOR-engaged endosomal ubiquitin network components that target the active receptor to lysosomes. Briefly, DOR-APEX2-expressing cells were exposed to the opioid agonist for varying periods of time. Biotinylated proteins identified from each time point were quantified against proteins identified from spatial references using label-free quantification. This experiment led to the identification of 29 specific interacting proteins, which were classified as being biotinylated at an early, middle, or late phase following receptor activation. WWP2 and TOM1, two ubiquitin-linked proteins that showed the strongest labeling in the late phase, were further validated as network components that mediate endosomal sorting of the DOR to the lysosome [89].

APEX has also been utilized to probe biased signaling. One example is the angiotensin II type 1 receptor (AT1R), which responds to the peptide hormone angiotensin II (Ang II) and has a critical role in cardiovascular physiology. AT1Rs are targeted for the treatment of cardiovascular diseases, and G-protein- and β-arrestin-biased AT1R ligands have been developed [76]. HEK293 stably expressing AT1R-APEX2 were treated with the full agonist Ang II, a partial agonist, two G-protein-biased agonists and two β-arrestin-biased agonists [91]. Biotin labeling was activated 1.5 min, 10 min, and 1 h after receptor activation. Controls included ligand-free cells with and without biotin labeling and cells treated with a receptor blocker. In order to compare the signaling landscape for all treatments at each time point, TMT-based quantitative proteomics was performed to enable parallel and quantitative analysis of the AT1R signaling landscape in all treatments through a single MS analysis. This study has provided novel functional insights into AT1R-biased signaling and ligand-dependent receptor trafficking patterns and kinetics. For instance, G-protein-biased ligands were found to be associated with receptor trafficking to endosomes, lysosomes, and clathrin-coated vesicles, while β-arrestin-biased ligands were associated with proteins involved in F-actin cytoskeleton remodeling, membrane ruffling, lamellipodium and, interestingly, the centrosome, suggesting the potential involvement of AT1R in cell cycle regulation [91].

To our knowledge, APEX-MS is currently the only platform that allows the identification and quantification of GPCR signaling components within a native condition in an unbiased, medium- to high-throughput fashion. However, the GPCR APEX is still in its infancy as, in all studies reported so far, the experiments were performed in a cell model system, with exogenous expression of the APEX-tagged receptor and treatment of the cells with high doses of agonists. This may result in protein interactions or signaling events that may not happen in more physiologically relevant settings. The next step is for such an experiment to be performed in primary cells and with tagged proteins expressed at the endogenous level. For instance, β2AR APEX could be performed in iPSC-derived cardiomyocytes expressing APEX-tagged receptors where the receptor function is of importance, and its interactome could be studied in parallel with β1AR APEX in normal and disease conditions. Moreover, the platform can also be utilized to study organelle-resident GPCRs. For instance, more than 40 functional nuclear GPCRs (nGPCRs) have been identified in several cell types, and there is a growing body of evidence supporting a pathological role for these receptors in the cardiovascular and nervous systems [81,93]. However, the exact signaling landscape and downstream targets of nGPCRs require further investigation. The APEX-MS could provide a platform to identify and discriminate, for instance, the nuclear receptor signaling components from cell membrane receptor signaling components, without the need to biochemically isolate the nucleus, paving the way for organelle-specific drug development.

### 3.2. A-Kinase Anchoring Proteins (AKAPs)

One of the key aspects of cAMP signaling compartmentation is the anchoring of PKA holoenzymes in proximity to the source of cAMP production and/or to their substrates, which is achieved via PKA binding to AKAPs [94]. Proteins in the AKAP family can roughly be divided into three categories: RI-specific AKAPs, RII-specific AKAPs and dual-specificity AKAPs, which can anchor both RI and RII, although with rather different affinities. The AKAP-PKA interaction is based on binding between an amphipathic α-helix in the AKAP and a complementary surface formed by a dimer of R subunits and involving the dimerization-docking domain located at the N terminus of R [95]. AKAPs also serve as scaffolds that assemble into the cAMP signaling complexes’ signal terminators, such as protein phosphatases and PDEs, as well as additional kinases and other signaling proteins.

Owing to the AKAPs’ diverse subcellular localization and ability to assemble multiple signaling components, customized cAMP signaling units can then be arranged at specific local sites. The functional importance of AKAPs in organizing cAMP local domains is supported by several meta-analysis studies that identify a link between SNPs and mutations in AKAPs with increased risk of diseases [96,97,98,99]. In one example, an SNP that results in one amino acid change from isoleucine to valine in the anchoring domain of AKAP10 was shown to reduce RI-binding affinity by 3-fold. The isoleucine variant was unable to target RI to mitochondria, resulting in RI accumulation in the cytosol [96]. In another example, the cardiac AKAP Yotiao, harboring a single mutation that disrupts its interaction with the PKA-modulated I(Ks) potassium channel, was demonstrated to eliminate the functional response of the channel to cAMP, causing long QT syndrome [100]. Alterations in AKAP expression levels and interactions have also been associated with pathologies, including heart failure [101], disorders of the nervous system [102] and male infertility [103].

Over the past decade, studies have evaluated selective targeting of individual AKAP signaling complexes as a strategy to manipulate cAMP-mediated cellular events in diseases [8], with the view that this approach would cause only ultra-fine changes at the relevant signaling complex without affecting other cellular functions. This line of investigations has driven the development of fundamental knowledge, especially of the architecture of individual PKA-AKAP signaling complexes, in physiologically relevant systems under normal and pathophysiological states. Proteomics offers a powerful set of tools for AKAP research, starting from the cAMP-centric proteome-wide screening of the cAMP interactome in a given cell or tissue to the identification of an AKAP’s anchoring landscape. The chemical proteomics platform based on cAMP affinity chromatography was developed to purify cAMP primary interactors (e.g., PKA isoforms, EPACs and PDEs) from a lysate of any origin. At the same time, secondary interactors (e.g., AKAPs, and PKA substrates) could be co-purified [104,105,106,107]. The enrichment of the co-purified proteins represents solely their association with R isoforms and not the expression levels, as in the case of primary interactors. The method makes use of different synthetic cAMP-analog resins that show different affinities toward PKA isoforms, PDEs and other primary interactors [108,109]. When combined with quantitative mass spectrometry, the method provides cues for profiling dynamic rearrangements of signaling scaffolds in response to stimulation or diseases [101,110,111]. For instance, cardiovascular conditions are often associated with inappropriate activation of blood platelets. An increase in intracellular cAMP was shown to interfere with platelet activation via PKA-mediated phosphorylation events impacting the organization of the platelet cytoskeleton [112]. Evaluation of the differential cAMP interaction landscape of resting and stimulated platelets revealed the cAMP signaling domain(s) involved in the inhibition of platelet activation. The cAMP interactome was enriched using cAMP resins from resting-state, and collagen-stimulated platelet lysates and quantitative LC-MS/MS analysis were conducted on differentially dimethyl-labeled peptides. The platform first identified the landscape of the cAMP interactome in platelets, consisting of three PKA isoforms (PKA-RIα, RIIα and RIIβ) and seven AKAPs. Stimulation led to increased anchoring of PKA-RII to AKAP2 and AKAP9 [110], suggesting the involvement of these PKA-AKAP pools in the inhibition of platelet activation. In another example, the same platform was used to investigate altered PKA signaling complexes in heart failure (HF) [101]. The study design involved a comparison of enriched cAMP interactors from normal and end-stage failing human heart tissues in a label-free fashion. Since R subunits were found to be down-regulated in failing hearts, enriched AKAPs were then normalized against their preferred R isoform before comparison to the healthy control. The study revealed extensive reorganization of PKA associations to AKAPs and their substrates. In failing hearts, a much larger population of R was found to associate with AKAP18 γ/δ, PALM2-AKAP2 and SPHKAP. Additionally, an increase in MAP2-RII and a decrease in Yotiao-RII interactions were observed. This type of information may offer a starting point for the development of specific therapeutics to locally rescue dysfunctional signaling at specific scaffolds. The analysis also revealed a decrease in the enrichment of several myofibrillar PKA targets (e.g., troponin I, titin and cardiac myosin binding protein C) from the patient materials, suggesting an uncoupling of the myofibril pool of PKA from their substrates [113].

The cAMP-centric chemical proteomics screening approach can provide insight into which PKA-AKAP scaffolds might underlie the physiological process or pathological condition of interest. To further characterize the molecular composition of a relevant cAMP signaling domain, interaction proteomics can be applied by using the scaffold protein as bait. For example, SPHKAP was among the AKAPs that showed a dramatic increase in association with PKA in HF [101], suggesting an important role for this AKAP in cardiac pathophysiology. SPHKAP was discovered by cAMP affinity chromatography as a novel and highly abundant AKAP in the heart [104]. Later, its affinity to RI was established, making it the first RI-specific AKAP ever identified in mammalian species [114]. To gain more insight into its function, AP-MS was employed to reveal the SPHAKP anchoring landscape. Anti-flag affinity resins were used to capture flag-SPHAKP interacting proteins in HEK293 cells expressing flag-SPHKAP. Enriched proteins were identified by LC-MS/MS and compared, by a spectral counting method, to the proteins identified from the control experiment performed using empty vector-transfected cells. Strongly enriched with SPHKAP were RI and several members of the MICOS, a mitochondrial inner membrane complex involved in the formation and maintenance of mitochondrial cristae, and several mitochondrial outer membrane proteins that together form the mitochondrial intermembrane space bridging (MIB) complex [115]. The detection of SPHKAP in the mitochondrial intermembrane space in cardiac myocytes and the identification of MICOS proteins as PKA substrates [115,116] indicate that SPHKAP scaffolds RI at the MICOS/ cristae domain.

AKAPs contain targeting domains to anchor PKA to distinct subcellular compartments and in proximity to specific substrates. In different cell types, the same AKAP can be found at different cellular locations, engaging with different sets of proteins to serve different functions. For instance, in cardiomyocytes, AKAP18γ/δ localizes to the sarcoplasmic reticulum, where it scaffolds PKA with the Ca^2+^ pump SERCA-PLN complex to regulate adrenergic effects on Ca^2+^ re-uptake [117], while in renal collecting duct cells it targets PKA to aquaporin 2 (AQP2)-bearing vesicles, to regulate shuttling of the water channel from the cytosol to the plasma membrane [118]. Endogenous AKAP18γ was found in both the nucleus and cytoplasm of oocytes [119]. However, the exact function of the cytosolic and nuclear pools of the AKAP is unclear. Mapping the anchoring landscape of specific AKAPs is a valid approach to defining the specific function of a specific cAMP signaling domain. As a proof of concept, AKAP18γ was modified to localize either to the nucleus or the cytoplasm. The nuclear and cytosolic AKAP18γ were tagged with miniTurbo biotin ligase, overexpressed in HEK293T cells, and BioID-MS was performed in a label-free fashion to identify and compare the proximal proteomes of the two AKAP18γ pools. The cytosolic pool of AKAP18γ was shown to associate with proteins involved in the cell cycle and regulation of translation, while the nuclear pool was specifically associated with the RNA splicing machinery [98].

Even though the identification of components of AKAP signaling scaffolds in cell models can yield insights into the potential cellular function regulated by that complex, one should bear in mind that this role may be cell-type and/or cellular-state specific. Thus, experiments performed using physiologically relevant cells, and tissues are recommended. For AKAPs where a specific antibody is available, the signaling complex can then be studied directly in an endogenous context. For instance, AKAP12 is required to promote endothelial cell migration through unknown mechanisms. In order to identify components of the AKAP12-associated cAMP signaling domain involved in the regulation of cell migration, immunoprecipitation was performed by incubating an AKAP12-specific antibody or control IgG with the lysate from human endothelial cells where migration had been triggered by treatment with vascular endothelial growth factor (VEGF). Enriched proteins were analyzed by LC-MS/MS, and comparison was performed using the LFQ approach. The study revealed that, in migrating endothelial cells, endogenous AKAP12 was strongly associated with multiple key regulators of actin dynamics and actin filament-based movement and the VEGF stimulation was translated, via PKA-mediated phosphorylation events, into actin cytoskeleton remodeling and cell movement [120].

As mentioned earlier, AKAPs scaffold PKA holoenzymes with their substrates. This has led to the concept of AKAP-centric phosphoproteomics profiling to identify compartmentalized PKA phosphorylation events orchestrated by a specific AKAP. The RII-specific AKAP Cypher/ Zasp was shown to be strongly associated with dilated cardiomyopathy (DCM), and the cardiac L-type CaV1.2 calcium channel was the only known PKA effector regulated by this AKAP upon β-adrenergic activation [121]. To screen for additional cardiac PKA substrates regulated by Cypher, neonatal cardiac myocytes from wild-type and Cypher-KO mice were treated with isoproterenol to trigger β-adrenergic/PKA phosphorylation events before cell harvesting. Phosphopeptides were enriched from cardiac tissue digests by the TiO_2_ enrichment method and identified by LC/MS-MS in a label-free fashion. Compared to WT mice, the study identified 216 phosphopeptides differentially expressed in the Cypher-KO tissue, about half of which were down-regulated. In this case, the hypophosphorylated peptides that harbor a PKA consensus motif are likely to represent PKA-dependent phosphorylation events regulated by Cypher. These include β-catenin (Ser675), vimentin (Ser72), and the known PKA substrate Troponin I (Ser23/24) [122]. The study led to several discoveries, including a novel role for Cypher in the modulation of β-catenin transcriptional activity and cardiomyocyte proliferation via β-catenin phosphorylation [122]. This discovery highlighted a signaling crosstalk between the Wnt/β-catenin and the cAMP signaling pathways, possibly mediated by the colocalization of Cypher, vimentin and integrin β1 at the costamere [122].

### 3.3. cAMP-Dependent Protein Kinase (PKA)

The principal intracellular target for cAMP is PKA. PKA activation requires cooperative binding of cAMP to two sites on each R subunit. Upon binding of four molecules of cAMP, the catalytic subunits are freed from the inhibitory action of R subunits and phosphorylate nearby substrates at serine/threonine residues located within the consensus amino acid sequence (R/K)-(R/K)-x-(pS/pT), where x is any amino acid [123,124].

Besides serving as a cAMP binding site, the R subunit can be considered as an adaptor that tethers bound C subunits to distinct subcellular sites near their protein targets. cAMP binding to R subunits is thought to release active C subunits, raising the question of how specificity is maintained when the active C subunit is no longer attached to the AKAP scaffold. A study in living neurons may provide cues to this conundrum [125]. Upon activation, only a fraction of Cα was dissociated from anchored holoenzymes. The freed Cα was preferentially translocated to the membrane via the myristoylation of its N-terminus, enabling phosphorylation of membrane-bound substrates [125]. However, how the C subunit is uncoupled from the membrane for signal termination remains unknown. Contrary to current dogma, recent studies in Scott’s laboratory indicate that local PKA action can proceed through intact holoenzymes [126]. In the case of the PKA type II-AKAP79 complex, a substantial proportion of anchored PKA was shown to remain intact and proximal, within a distance of 15–25 nm, to anchoring sites and substrates [126], indicating that C phosphorylation activity may be more locally confined than previously appreciated. Interestingly, the dimension of anchored PKA action measured in this study is consistent with the estimated dimension of cAMP gradients shaped by PDEs (10–30 nm) [9,10].

As a major target of cAMP, it is safe to assume that PKA resides in a large proportion of functional cAMP signaling units. In a subcellular or suborganellar compartment where the cAMP interactome is not completely known, PKA can thus be used as a “proximity bait” to discover the landscape of cAMP signaling domains in that compartment. For instance, it is apparent that there are two separate cAMP signaling systems that operate on the surface of and inside mitochondria. The signaling cascades hosted at the outer mitochondria membrane are well established and are involved in the regulation of apoptosis and mitochondrial dynamics [127,128]. In contrast, the secluded intramitochondrial cAMP signaling system is still debated [129,130], even though all the components necessary to form cAMP functional domains have been identified within the organelle, potentially enabling intramitochondrial cAMP production in response to CO_2_/HCO_3_ [129]. Although components of the electron transport chain and TCA cycle have been hypothesized to be phosphorylated by a mitochondrial pool of PKA [129], studies aiming at the characterization of the intramitochondrial cAMP signaling landscape remain scarce, partly due to the difficulty of generating pure mitochondrial sub-compartment fractions. New insight was recently provided by the application of BioID-MS to characterize the PKA proximal proteome in the mitochondrial matrix [131]. First, the authors confirmed that a pool of Cα subunits is present inside the mitochondria in HeLa cells. Cα proximal proteins were enriched from cells overexpressing fusion proteins of the matrix-localized Cα and BioID2 (mt-PKA-BioID2). Cells transfected with empty vectors were used as a control. After quantification by spectral counting, 33 of the enriched proteins (~15%) were found to be mitochondrial, of which around 60% were predicted to harbor a PKA phosphorylation motif. These proteins are involved in various mitochondrial processes such as stress response, TCA cycle, protein synthesis and degradation. Interestingly, around two-thirds of the mt-PKA potential targets identified overlap with interactors of the prohibitin complex, an inner mitochondrial membrane ring-like structure important for cristae morphogenesis and functional integrity of the organelle [131]. These proteins could represent the intramitochondrial cAMP signaling domains. However, the question remains whether these predicted PKA targets are phosphorylated by the mitochondrial pool of PKA. Quantitative PKA phosphorylation profiling in the conditions that specifically lead to intramitochondrial cAMP production could help resolve this point. In addition, no expression of free biotin ligase was used in this experiment as a control, making it difficult to draw firm conclusions on the specificity of the interactions detected. Despite these caveats, this study provides an example of how compartmentalized cAMP signaling can be explored at subcellular locations where the cAMP interactome is uncharacterized. This type of study could be further developed to include quantitative PL-MS experiments using R or C isoforms targeted to specific locations. The use of appropriate controls would allow mapping of the PKA proximal proteome, including AKAPs, PKA substrates, as well as other components residing in the local signaling domains.

A recent study from Scott’s laboratory demonstrated the use of this strategy to identify the local landscape of cAMP signaling events relevant to Cushing’s syndrome [132]. Somatic mutations L205R and W196R, found in the PRKACA gene encoding for PKA-Cα, were found to drive the overproduction of the stress hormone cortisol in this condition [132]. Mechanistically, the mutations were found to disrupt the binding of Cα to R subunits, resulting in the uncoupling of the C variants from AKAP scaffolds [132]. In order to gain insight into the new landscape associated with the disanchored C variants, the proximal proteomes enriched from adrenal cell lines stably expressing low level of a miniTurbo-tagged version of the two Cα mutants were compared with the proteome of wild-type C using an LFQ approach. Besides the expected reduced association with R and AKAPs, both variants revealed enrichment in different nuclear domains, including nuclear pore, inner nuclear membrane, spliceosomal complex, and histone modification, in agreement with imaging data showing a more pronounced nuclear localization of the C variants. In a general sense, a mislocalized active kinase can lead to non-physiological phosphorylation events, driving pathological consequences. In order to gain further insight into aberrant phosphorylation induced by the two mutations in Cα, phosphopeptides were enriched from each variant proximal proteome using an Fe-IMAC approach and compared with those of the WT enzyme. Apart from the expected decrease in phosphorylation in AKAPs, the two variants appeared to engage with different substrates and potentiate distinct downstream mitogenic signaling pathways [132]. Whether the abnormal phosphorylation pattern observed drives the pathological processes in Cushing’s syndrome remains to be established. Overall, this study highlights the importance of signaling compartmentalization and the power of proteomics approaches to help unravel aberrant signaling events.

In a different paradigm, a comprehensive analysis of phosphorylation changes upon inhibition of PKA activity can uncover specific cellular processes selectively regulated by PKA. System-level phosphoproteomic profiling of a kinase-specific phosphorylation activity can be considered a large-scale proximity analysis where the kinase comes into physical contact with its substrates [133]. As a proof of concept, Knepper’s laboratory established a double Cα and Cβ knockout vasopressin-responsive mpkCCD cell lines, which they used as a collecting duct (CD) principal cell model to study vasopressin signaling [134]. Vasopressin binds to the vasopressin receptor (V2R), leading to an increase in cAMP production and PKA activation. The actions of vasopressin are associated with several physiological responses, including the regulation of the water channel aquaporin-2 (AQP2) to modulate osmotic water permeability in CD epithelia [133]. To identify vasopressin-mediated cellular processes that are regulated by PKA phosphorylation, the authors first identified PKA-dependent phosphorylation events in PKA-C KO cells. Phosphopeptides were enriched from light SILAC-labeled PKA-C KO and heavy SILAC-labeled PKA-intact mpkCCD cell lysates. Differentially regulated phosphorylation sites were classified according to the sequences surrounding the phosphorylated amino acids. As expected, down-regulated phosphorylation sites in double KO cells were predominantly PKA sites, although other basophilic kinase sites were also identified. Interestingly, most of the up-regulated sites were proline-directed kinase sites, suggesting that in CD cells, proline-directed kinases, such as MAP kinases, are negatively regulated by PKA, either directly or indirectly. The authors further mapped the differentially regulated phosphorylation events to known vasopressin-regulated physiological responses to build a PKA-dependent signaling landscape in response to vasopressin. Several key cellular processes were identified as PKA-regulated. These include transcriptional regulation of the AQP2 gene, AQP2 phosphorylation and translocation to the apical plasma membrane, AQP2 exocytosis, MAP kinase signaling, and actin dynamics [134].

The authors further investigated the PKA-dependent signaling landscape specific to Cα and Cβ. In native CD and mpkCCD cells, the two catalytic subunits are expressed at a comparable level, in keeping with the notion that they are functionally non-redundant [135]. To dissect the phosphorylation landscape associated with each catalytic subunit, Cα or Cβ knockout mpkCCD cell lines were generated [136]. Phosphopeptides were enriched from TMT-labeled knockout and PKA-intact cell lysates using a TiO_2_ and Fe-IMAC sequential enrichment method. As expected, the two kinases were shown to have a substantially different set of phosphorylation targets, with most of the PKA sites being found to be decreased in Cα-null cells. Interestingly, several differentially-regulated sites of PKA-RI and RII, AKAPs, and PDEs were identified in Cα-null cells, while only one site in AKAP12 was identified in Cβ null cells, suggesting that Cβ is less associated with PKA-AKAP complexes than Cα is. Cα targets were mainly associated with cell membranes and vesicles, whereas Cβ targets were related to the actin cytoskeleton and cell junctions [136].

If these findings are taken together, they indicate that the two catalytic subunits operate at discrete locations, engaging with distinct sets of local targets. Cα might be associated with a cAMP signaling domain in the AQP2 storage vesicles or at the apical membrane, while Cβ may operate at actin barrier domains. One can then utilize these two catalytic subunits as bait to dissect a specific signaling pathway in response to vasopressin. For example, a major element of vasopressin action in CD cells is the dramatic rearrangement of the actin barrier for proper trafficking of AQP2-bearing vesicles to the apical membrane [137]. The local signaling events that regulate actin barrier dynamics are a critical yet incompletely understood aspect of water homeostasis. To dissect the landscape of local cAMP/ PKA signaling involved in actin barrier remodeling, one could utilize Cβ as a bait protein for PL-MS analysis to identify the local signaling complexes/PKA effectors in the presence and absence of vasopressin stimulation. To further dissect the local cAMP domains specific to a specific stimulus, the list of PKA-dependent phosphorylation targets identified from such studies could be further examined in combination with the list of phosphorylation events that are differentially regulated in response to a ligand of interest (e.g., vasopressin, in the case of CD cell lines [138,139]).

### 3.4. Phosphodiesterases (PDEs)

The level of cAMP is determined by the rate of synthesis by ACs and the rate of degradation by the cyclic nucleotide-hydrolyzing enzyme PDEs. There are eight different families of cAMP-degrading PDEs (PDE1, 2, 3, 4, 7, 8, 10, 11). PDE families include multiple genes and several splice variants, giving rise to multiple isoforms. PDE isoforms differ greatly in their tissue distribution, and multiple PDE isoforms can be expressed in an individual cell type [140]. Different PDE isoforms can be targeted to different subcellular compartments by targeting sequences or via interactions with local macromolecular complexes. Hydrolysis of cAMP is the only major mechanism to reduce the second messenger concentration, and by degrading cAMP to a different extent at different sites, PDEs play key roles in shaping and maintaining cAMP signaling compartmentalization. In fact, a large body of work demonstrates the role of PDEs in organizing cAMP signaling domains [141,142,143]. PDE enzymatic activity and PDE clustering, together with cAMP buffering, have been identified as key factors that contribute to the formation of local cAMP domains [10,144,145]. Using fluorescence resonance energy transfer (FRET)-imaging, different PDEs were demonstrated to organize differently scaled domains of nanometer size [10]. In such domains at basal state, a cluster of PDEs was shown to function as a local sink by actively degrading cAMP in their immediate vicinity. When cAMP production increases upon GPCR stimulation, the local PDE capacity is saturated, leading to local PKA activation and downstream signaling [10]. In another FRET-based study, it was observed that the cAMP synthesizes on stimulation of GPCR with low doses of agonist remains confined in the vicinity of the receptor thanks to the activity of local PDEs, generating a receptor-associated independent cAMP nanodomain, or RAIN, where anchored PKAs is strongly activated by a high concentration of local cAMP [9]. 

PDE inhibitors are currently in use to treat conditions that associate with the dysregulation of cAMP signaling, owing to the role of PDEs as signal terminators. For example, PDE3 inhibitors are used clinically for the treatment of acute, refractory heart failure. However, these drugs are associated with significant side effects [146,147], possibly due to their non-selective disruption of PDE3 activity in all domains where isoforms of this family operate, some of which could be vital for normal physiological function. In such a scenario, therapeutic strategies that specifically target only the unique pool of PDE isoforms that control the desired function would be ideal. For instance, displacement of a resident PDE from a local cAMP signaling compartment would result in a local increase in cAMP with restricted activation of PKAs only in that compartment and exclusive phosphorylation of neighboring targets. However, to be able to exploit compartment-selective displacement of individual PDEs for therapeutic purposes, a blueprint of which cAMP signaling compartments are regulated by which PDE isoform and a detailed understanding of how the PDE is retained in that compartment are required.

PDE-selective inhibitors have been exploited to study the roles of different PDEs in the regulation of cyclic nucleotide signaling. A collection of inhibitors highly selective for each PDE family is commercially available [87]. These inhibitors can be used in combination with quantitative phosphoproteomics to interrogate phosphorylation events regulated by each PDE family. For example, work by Ong’s and Beavo’s laboratories used PDE-centric profiling of cAMP-mediated phosphorylation after treatment with family-selective PDE inhibitors, with or without a receptor agonist co-stimulation, to increase local cAMP signals and generate changes in phosphorylation events on the target proteins in the vicinity of the PDE [71,72,148].

This platform was used to profile cAMP-mediated phosphorylation events involved in steroidogenesis in the mouse cell line MA-10 [71]. An increase in cAMP levels upon lutropin-choriogonadotropin hormone receptor (LHCGR) activation is known to induce steroidogenesis, although the cAMP signaling events involved in this regulation are not fully understood. It was found that in MA-10 cells, only the co-inhibition of PDE4 and PDE8 was able to increase the intracellular cAMP level as observed after LHCGR activation, even though to a lesser extent [71]. In order to investigate cAMP signaling events in a condition that mimics LHCGR receptor activation, two overlapping sets of experiments were carried out in triple SILAC-labeled MA-10 cells. The first set included individual SILAC (light, medium, heavy) labeled cells treated with DMSO control, PDE4 inhibitor and a combination of PDE4 and PDE8 inhibitors. The second set was designed to compare DMSO control, PDE8 inhibition, and co-inhibition. For each experiment, individual SILAC-labeled lysates were combined, digested, and phosphopeptides were enriched using an Fe-IMAC approach. The study reported a rich dataset, including the identification of ~28,000 phosphorylation sites in ~14,000 different phosphopeptides derived from ~5000 proteins. Among the regulated sites, ~23% were identified as PKA sites, suggesting that multiple kinase-dependent pathways acted downstream of PKA. This study demonstrates how the identification of cAMP-mediated phosphorylation events can be used to construct an atlas of (PDE-regulated) local PKA activity, which, in the case of MA-10 cells, includes a great number of functional compartments involved in the regulation of cell cycle, insulin receptor signaling, transcription, endocytosis, vesicle trafficking and others. In addition, the identification of unique PDE8-regulated phosphorylation sites suggested the distinct role of this PDE in apoptosis signaling.

In another study, this quantitative phosphoproteomics platform was used to characterize, in label-free experiments, the PDE-regulated phosphoproteomes in the T lymphocyte cell line Jurkat treated with various combinations of PDE inhibitors and a low concentration of prostaglandin 2 [72]. Treatment with individual PDE inhibitors was not sufficient to induce changes in total cAMP or phosphorylation events, and simultaneous inhibition of multiple PDEs was required to obtain such a change. Different combinations of PDE inhibitors were found to impinge on distinct pools of cAMP, in turn regulating different cellular functions. For example, several functional compartments involved in the regulation of RNA metabolism were controlled by PDE1, 7, and 8, while the nuclear domains involved in histone trimethylation and DNA repair were modulated by PDE3 and 4 [72]. One of the key advantages of using this platform lies in the unbiased identification of regulated phosphorylation events and the sheer number of phosphorylation sites identified. Potentially, this platform can be utilized in preclinical screening to determine the possible collateral adverse effects that might be originated from the inhibition of PDE(s).

One limitation of the PDE family-centric phosphoproteomics profiling described above is that it does not provide insight into the specific role of individual PDE isoforms, as there are no available isoform-selective PDE inhibitors. The analysis of PDE isoform-selective interactomes, however, can overcome this shortcoming, and by cross-referencing the PDE family–dependent phosphoproteome with the PDE isoform-specific interactome, a more detailed view of the full cellular landscape of cAMP signaling events in response to specific receptor activation should be attainable. This information could then be used as a starting point to design single-domain targeting interventions, e.g., based on individual PDE isoforms displacement, to raise cAMP only at specific locations, resulting in highly targeted functional effects.

Surprisingly, data on a PDE isoform-selective interactome are very scarce. The analysis of the interactions in which a PDE isoform is involved is critical to link a specific enzyme with specific cellular responses. For example, treatment with an AC agonist, ectopic expression of PKA Cα, or pharmacological inhibition of endogenous PDE2A activity were shown to impair translocation of the ubiquitin ligase Parkin, a key effector of mitophagy to depolarized mitochondria, hence inhibiting mitophagy [149,150]. Of the three PDE2A isoforms, only isoform 2 (PDE2A2) is localized to mitochondria. In an attempt to define the mitochondrial target(s) involved in the PDE2A2/cAMP/PKA-dependent regulation of mitophagy, we performed a strep-tag-based AP-MS analysis to identify the interactome of PDE2A2 (Figure 5) [149]. Enriched proteins from strep-tagged PDE2A2 expressing cells were quantified against those from control cells expressing an irrelevant protein containing strep tag by the LFQ method. As expected, we found multiple mitochondria-resident proteins enriched in the PDE2A2 pull-down. The PDE2A2 interactome included the MIB complex (Figure 5), a core component of which is mitofilin (IMMT), which had been previously identified as a PKA target [116]. PKA phosphorylation of mitofilin was shown to be involved in Parkin recruitment to damaged mitochondria [116]. The study further demonstrated that PDE2A2 regulates a pool of mitochondrial cAMP that enables PKA-dependent phosphorylation of mitofilin, thus promoting Parkin recruitment and mitophagy. The study also demonstrates, as a proof of concept, that the analysis of PDE isoform-specific interactomes can effectively identify the isoform-associated cAMP signaling compartments. Interestingly, the analysis also revealed the enrichment of several members of the endosomal sorting complex required for transport (ESCRT) and autophagosome formation (Figure 5). The future challenge will be to establish whether distinct pools of PDE2A2 participate in the different complexes and to resolve the PDE2A2-regulated phosphorylation events and cellular targets involved in the ESCRT-dependent and autophagy processes. However, this study clearly demonstrates that individual PDE isoforms can be involved in the organization and regulation of multiple cAMP signaling domains at different locales within the same cell.

## 4. Conclusions and Perspectives

As more information becomes available on how cAMP/PKA signaling exploits compartmentalization to translate individual signal inputs into the appropriate functional outputs and how such organization is disrupted in disease, it also becomes apparent that the highly structured architecture of this signaling pathway provides extraordinary potential for the development of targeted interventions to normalize signaling only where required, without perturbing the entire network. The ability of PL-MS techniques to capture both stable, weak and/or transient PPIs in combination with state-of-the-art phosphoproteomics profiling technologies offers a powerful toolbox for resolving in detail the map of the cellular cAMP signaling landscape, both in physiological and pathological conditions. For example, PKA-R and/or PKA-C PL-MS can be performed to uncover the full repertoire of PKA signaling domains in the cell overall (Figure 6A). For a more targeted study, AKAP-centric and PDE isoform-centric PL-MS can be performed to gain more detailed insights into the subcellular distribution and regulation of individual domains (Figure 6B,C). Downstream signaling events regulated by specific AKAP or PDE can be identified by quantitative phosphoproteomics profiling in combination with specific AKAP gene knockout, use of AKAP-PKA disrupting peptides or PDE-family inhibitors (Figure 6D). Changes in phosphorylation events determined in these studies pinpoint the protein targets that can be mapped to specific domains and provide insight into the cellular function under the control of that specific domain. The information obtained from the analysis of the interactome and of the phosphoproteome can be combined to draw robust conclusions on the composition and function of individual cAMP signaling domains.

The map of the cAMP signaling landscape generated in this way can then be used to reveal potential protein targets or protein interactions that can be targeted to manipulate local signaling via domain-specific intervention. Several strategies have been proposed to intervene in discrete cAMP domains. Synthetic peptides mimicking the A-kinase binding domain of AKAPs have been developed to displace PKA from the AKAP scaffold [150,151,152]. These disrupting peptides are usually PKA-R isoform selective since they are all designed to interact with the same surface on the D/D domain of either RI or RII. These peptides represent useful tools to selectively probe anchored PKA signaling events in the laboratory [153] but have limited use as therapeutics since they interfere with multiple PKA-AKAP complexes. Knockdown of specific AKAPs to disrupt a specific signalosome may lead to more extensive effects than those desired as not only the PKA but also other signaling molecules brought together to the domain are also displaced (Figure 6D, blue box). A more selective strategy would be to locally modulate cAMP signals at specific domains by targeting the local PDE. Generally, PDE isoforms localize to different subcellular locations by protein interaction with AKAPs or other domain components. With the cAMP signaling map at hand, one can design a targeted intervention to increase cAMP signals/ phosphorylation of target proteins by displacement of local PDEs from specific signaling domains using disrupting peptides (Figure 6E, yellow box). This strategy has been validated in experiments where a synthetic peptide encompassing a stretch of amino acids in PDE4D5 involved in its binding to HSP20 was used to disrupt the PDE4D5/HSP20 interaction, leading to an increase in PKA phosphorylation of HSP20 and attenuated β-agonist-induced hypertrophy in neonatal cardiac myocytes [154].

Future studies will no doubt provide further details on the molecular composition, functional role and regulation of the multiple cAMP signaling domains in specific cell types and on how the cAMP signaling landscape is remodeled in disease. The already rich proteomics toolbox will see new developments and will significantly contribute to progress in this field, ultimately offering rational ground for the design of therapeutics that target molecular mechanisms with subcellular precision.

## Figures and Tables

**Figure 1 ijms-24-04667-f001:**
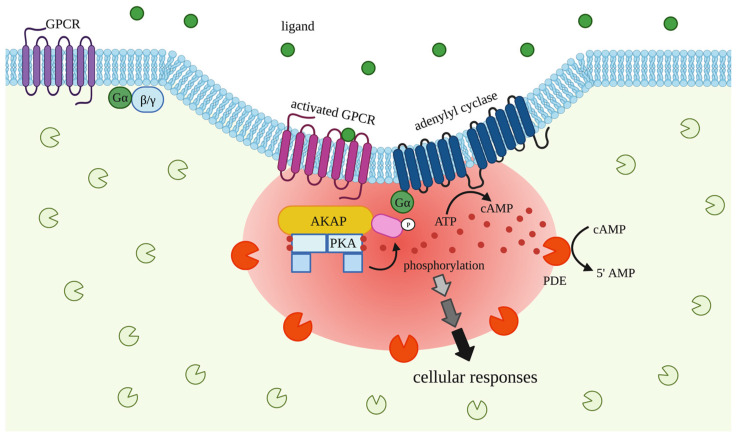
cAMP signaling domain at the plasma membrane. The schematic shows the key components of cAMP signaling, starting from a ligand binding to a specific GPCR at the plasma membrane. Receptor-ligand binding triggers a conformational change in the receptor (light purple), leading to GPCR-G protein interaction, which ultimately induces the release of α subunit of G protein (Gα) and production of cAMP by the enzyme adenylyl cyclase (blue). The local cAMP pool generated on activation of AC is represented as the red-shaded oval. The specificity in the translation of the message is ensured by the spatial constrain of the cAMP signal and by the scaffolding protein AKAP that tethers the major cAMP effector, PKA, to the site of cAMP production. Anchored PKA subsets are activated by a high concentration of cAMP to specifically phosphorylate only substrates localized in the vicinity. The amplitude and duration of the cAMP signal are shaped by the domain-associated PDEs (red Pac-Man symbols). Other cAMP compartments remain in the basal state, and their cAMP level is maintained at a low level by different pools of PDEs (light green Pac-Man symbols). The figure was created with Biorender.com.

**Figure 3 ijms-24-04667-f003:**
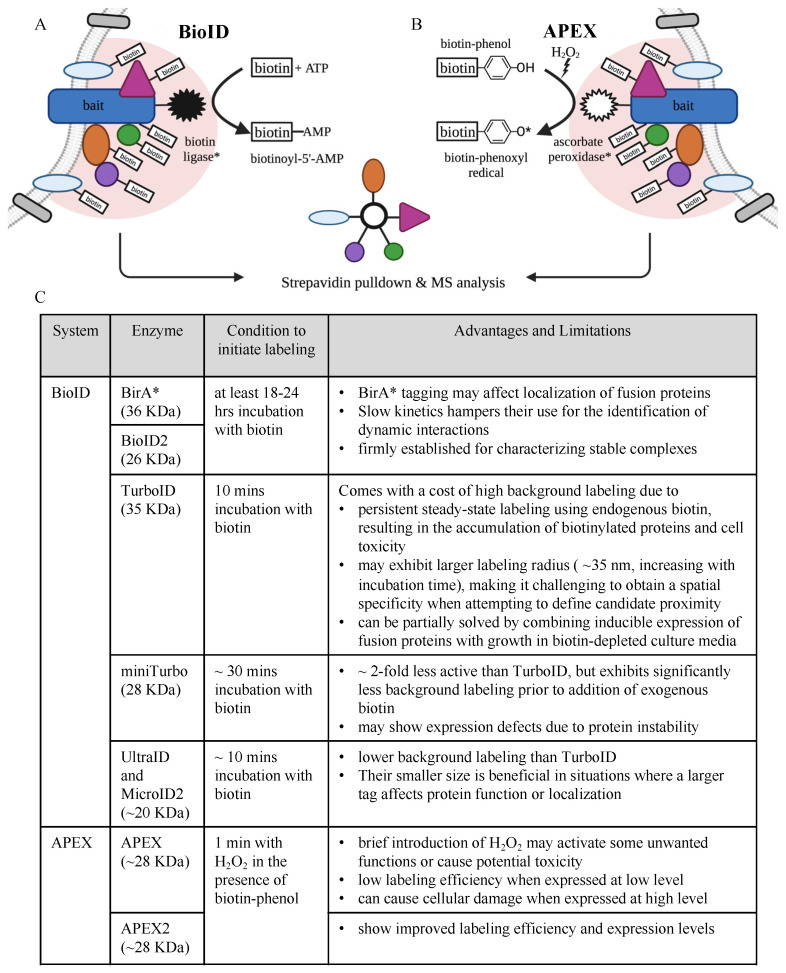
Schematic representation of PL-MS systems. (**A**) In BioID, a target (bait) protein is fused to a modified biotin ligase that catalyzes the conversion of biotin to biotinoyl-5′-AMP. This highly reactive form of biotin covalently attaches to accessible lysine residues within the labeling radius (depicted as a pink semi-circle) of ~10 nm. (**B**) APEX is based on the expression of bait protein fusion to the ascorbate peroxidase derivative that catalyzes the oxidation of biotin-phenol to reactive biotin-phenoxyl radical in the presence of H_2_O_2_. The radical reacts with electron-rich amino acids within a 20 nm radius, resulting in the biotin labeling of proximal proteins. Cells are lysed, and biotinylated proteins are enriched using streptavidin beads and analyzed by LC-MS/MS. (**C**) Summary of biotin labeling enzymes developed for BioID- and APEX-based PL-MS [43,44,45,46,47,48,49,53]. The figure was created with Biorender.com.

**Figure 4 ijms-24-04667-f004:**
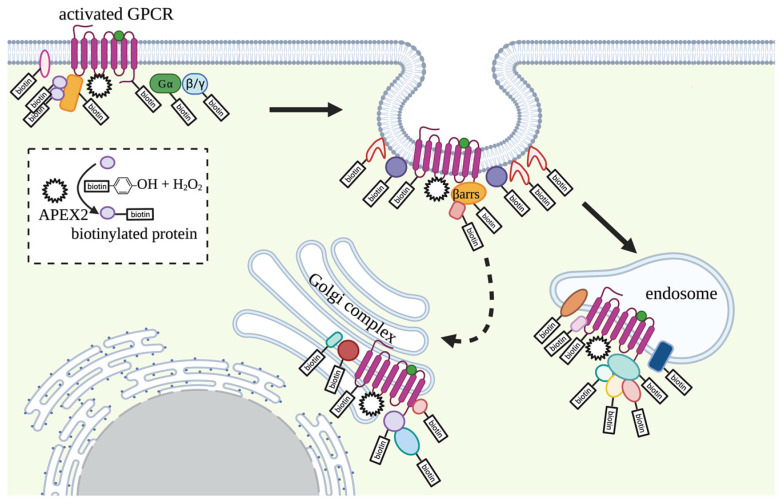
APEX-based PL-MS can capture the spatiotemporal dynamics of a GPCR signaling landscape. A GPCR fused to APEX2 is expressed in the relevant cell. After the addition of ligand and biotin-phenol, the ligand-bound GPCR-APEX2 can undergo dynamic and ligand-specific membrane trafficking to multiple membrane locations, such as endosome or the Golgi complex, where the receptor can engage with distinct signaling elements and/or sorting machinery. The addition of H_2_O_2_ at different time points after agonist stimulation initiates biotin labeling and generates time-resolved snapshots of GPCR signaling networks along the trafficking route. Biotinylated proteins are enriched and analyzed by LC-MS/MS to identify the compositions of the receptor local networks. Figure created with Biorender.com.

**Figure 5 ijms-24-04667-f005:**
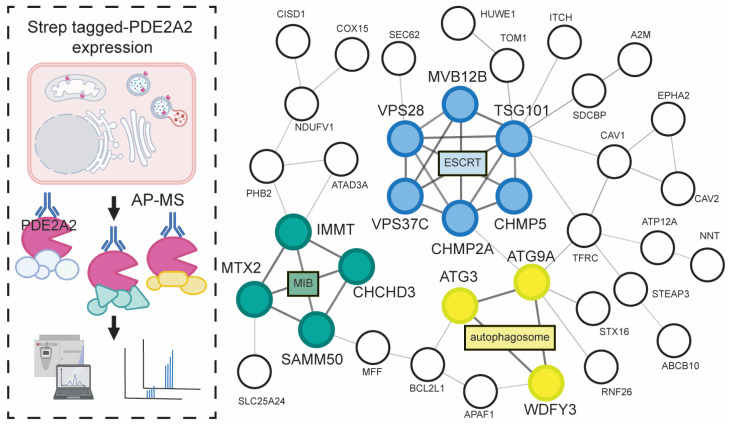
Resolving the PDE isoform-specific interactome. Recombinant-tag fused to a specific PDE isoform, in this example PDE2A2, is ectopically expressed. PDE isoform-associated complexes are enriched using a specific antibody to the recombinant tag and subjected to LC-MS/MS analysis. Various quantitative strategies can be combined with AP-MS to reveal specific interacting protein candidates. Interaction network analysis is performed to visualize protein complexes, hence cellular processes, that are potentially regulated by the PDE isoform. The network shown represents a fraction of the PDE2A2 interactome (primary data from [149]).

**Figure 6 ijms-24-04667-f006:**
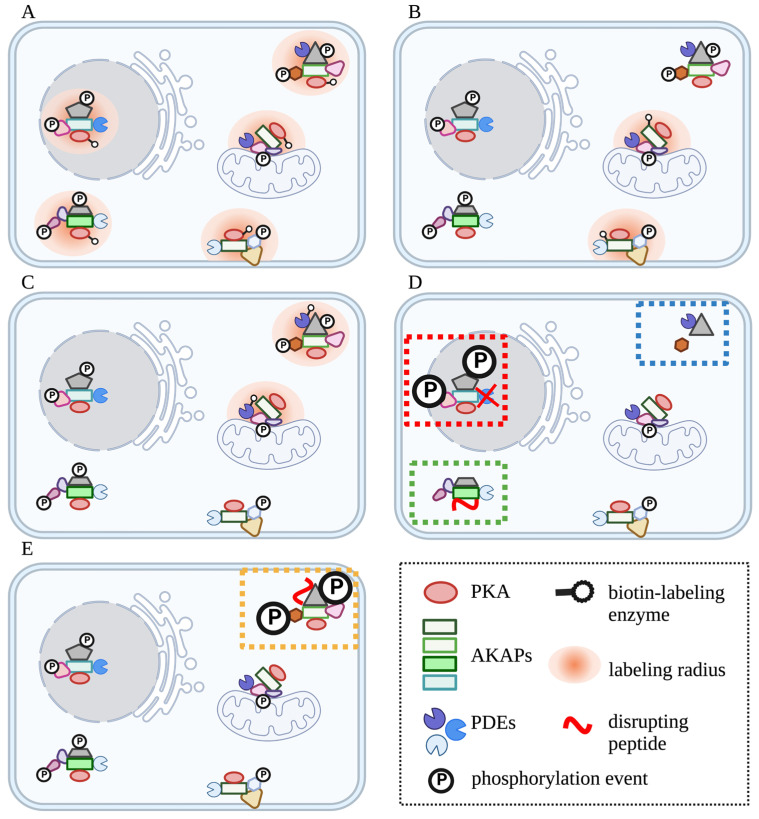
A multi-angular proteomics view on compartmentalized cAMP signaling. Illustration of (**A**) PKA-centric, (**B**) AKAP-centric, and (**C**) PDE-centric PL-MS strategies. For domain components-discovery screens, PKA-R or PKA-C can be used as bait. A fusion protein of bait-biotin labeling enzyme is expressed in the cell. Once biotin is added, PKA proximal proteins, including AKAPs, PDEs, and their substrates, are labeled. These can be enriched and identified by LC-MS/MS. For a more targeted study, AKAP or PDE isoform can be used. (**D**) For the identification of signaling events regulated by the AKAP or PDE, quantitative phosphoproteomics profiling in combination with AKAP gene knockout (blue box), AKAP-PKA disrupting peptides (green box), or PDE-family inhibitors (red box) can be used to identify changes in phosphorylation events upon disruption or perturbation of the domains. Interactome and phosphoproteome data can be combined to establish a reliable blueprint of individual cAMP signaling domains, including details of their molecular components and downstream signaling events. (**E**) A peptide displacement strategy involving a short peptide specifically designed to block the interaction between the local PDE and the protein component that retains the PDE within the domain. With the displacement of the PDE, the level of cAMP increases selectively within that particular domain and phosphorylation events are enhanced selectively on the target proteins contained within the domain (yellow box).

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
