# Peer review of "Using the Proteomics Toolbox to Resolve Topology and Dynamics of Compartmentalized cAMP Signaling"

_ijms, 2023, doi:10.3390/ijms24054667_

Round 1
Reviewer 1 Report
Review report
The review titled: “Using the proteomics toolbox to resolve topology and dynamics of compartmentalized cAMP signalling” examines cAMP compartmentalization which ensure the formation of local signalling domains in which are clustered together cAMP signalling effectors, regulators and targets involved in a specific cellular response. The focus in this review is on the use of proteomics toolbox for identification of the molecular components of the above domains in order to define the dynamic of cellular cAMP signalling and compiling the obtained data of the cAMP signalling in physiological and pathological conditions.
Introduction
-The authors comprehensively describe the role of cAMP and how it is obtained, as well as the various participants in cAMP signaling. Тhe new opportunities that allow cAMP compartmentalization to be exploited for for therapeutic interventions are well argued and would provide invaluable insight for targeted drug design.
-Further in the review successively follow descriptions and analysis of the principles of quantitative proteomics; Protein interaction and proximity profiling – basic principlesin which Several MS-based interactomics strategies are included; An overview of Phosphoproteomics; Application of proteomics methodologies to study compartmentalized cAMP signalling including the basic players of cAMP sygnalling and the Targeted manipulation of localised signalling events defined as a new strategy for precision medicine.
-In all parts of the review is followed a well-chosen strategy including the advantages, disadvantages of the used approaches, followed by examples and references for practical application.
-A part named conclusion is missing in the review. My suggestion is that the last part titled “Targeted manipulation of localised signalling events: a new strategy for precision medicine?“ can be reorganized in Conclusion where can be summarized the most important findings in the review together with the indicated perspectives revealing how cAMP signalling landscape is remodelled in disease.
-The included figures of excellent quality also help the good description and perception of the review.
Author Response
We feel that our final section adequately summarizes the content of the review and provides an outline of how the approaches described can impact understanding of physiopathology and development of new treatments and how the field is likely to develop in the future. We have therefore changed the title of the last section to “Conclusions and perspectives”.
Reviewer 2 Report
The authors explore the complex nature of the compartimentialisation of the cAMP signalling, with focus on the various proteomic methodologies available now. I have only one comment.
line 141> The detergents used for AP-MS of membrane proteins are typically non-ionic detergents therefore they do not disrupt PPIs.
Author Response
We would like to thank the reviewer for pointing this out. We realize that non-ionic detergents typically have little impact on PPIs, particularly if low concentrations are used. However, we would like to refer to the publication (DOI: 10.1021/acs.jproteome.7b00599), now included as a reference in our manuscript, that describes the effect of several commonly used (non-ionic) detergents on membrane protein complex solubilization and coimmunoprecipitation. We have now rephrased the sentence on line 141 to make this clearer.
Reviewer 3 Report
In this manuscript, Kovanich et al. provide a comprehensive review of the proteomics methodologies used for identifying the molecular components of the cAMP signaling pathway. The paper includes a general introduction to the cAMP signaling pathway and proteomic toolbox, followed by applying the toolbox in studying the cAMP signaling pathway. In general, the paper is well-organized and well-written. The information provided is complete and up-to-date. The only weakness is that, as a review paper, the manuscript is too long and could be improved a little bit for conciseness.
Author Response
Thank you for the comment. Our intention with this review is to give extensive information on the quantitative proteomics methodologies that are currently being applied to study cAMP compartmentalization in a way that is accessible to the non-specialist but still provides sufficient detail to enable critical analysis of the literature. We however take the point raised by this reviewer and have tried to reduce the length of some paragraphs and have removed sentences that were redundant. The text deleted is indicated in the tracked changes file.